# Microfluidic Array Enables Rapid Testing of Natural Compounds Against *Xylella fastidiosa*

**DOI:** 10.3390/plants14060872

**Published:** 2025-03-11

**Authors:** Francesca Costantini, Erica Cesari, Nicola Lovecchio, Marco Scortichini, Valeria Scala, Stefania Loreti, Nicoletta Pucci

**Affiliations:** 1Research Centre for Plant Protection and Certification, Council for Agricultural Research and Economics (CREA-DC), 00156 Rome, Italy; erica.cesari@crea.gov.it (E.C.); valeria.scala@crea.gov.it (V.S.); stefania.loreti@crea.gov.it (S.L.); nicoletta.pucci@crea.gov.it (N.P.); 2Department of Information Engineering, Electronics and Telecommunications, Sapienza University of Rome, 00184 Rome, Italy; nicola.lovecchio@uniroma1.it; 3Research Centre for Olive, Fruit and Citrus Crops, Council for Agricultural Research and Economics (CREA-OFA), 00134 Rome, Italy; mscortichini@yahoo.it

**Keywords:** *Xylella fastidiosa*, planktonic growth, biofilm formation, microfluidics, antibacterial activity

## Abstract

The bacterial pathogen *Xylella fastidiosa* (*Xf*), which causes several plant diseases with significant economic impacts on agriculture and the environment, remains a challenge to manage due to its wide host range. This study investigated the *in vitro* antibacterial effects of natural compounds, including *Trametes versicolor* extract, clove essential oil, and the resistance inducer Fossil^Ⓡ^, against *X. fastidiosa* subsp. *fastidiosa* using an antibacterial susceptibility testing (AST) method based on microfluidic channels. This novel method was compared with the traditional broth macrodilution method to assess its reliability and the potential advantages microfluidics offers. For each substance and test, both the ability to limit planktonic growth (reported as the minimum inhibitory concentration) and the ability to inhibit biofilm formation were evaluated. The results suggest that compared to the macrodilution method, microfluidic channels allow for a more rapid AST execution, use less material, and allow for real-time observation of bacterial behavior under a continuous flow of nutrients and antibacterial substances. All tested products demonstrated high antibacterial efficacy against *Xf* with the macrodilution method, yielding comparable results with microfluidic AST. These findings highlight the antimicrobial properties of the tested substances and establish the groundwork for applying this new technique to select promising eco-friendly products for potential future field applications in controlling *Xf*.

## 1. Introduction

*Xylella fastidiosa* (*Xf*) is a Gram-negative bacterium belonging to the *Lysobacteraceae* (formerly *Xanthomonadaceae*) family that causes serious disease in several economically significant agricultural crops. Pierce’s disease of grapevines (PD), Citrus Variegated Chlorosis (CVC), and Phony Peach Disease (PPD) are the most relevant diseases caused by *Xf* [1,2], as well as Olive Quick Decline Syndrome (OQDS), which is seriously affecting olive groves in the Apulia region (Italy) [3]. *Xf* is a xylem-limited bacterium vectored by xylem-sap-feeding insects, including sharpshooters, leafhoppers, and spittlebugs [4,5].

Among the research objectives aimed at limiting the spread of *Xf* diseases, pathogen control is a key focus. Targeting the bacteria involves testing various compounds both *in vitro* and/or in the field. Chemical control used as a crop protection strategy against phytopathogenic agents is proven to exhibit several risks related to toxicity. Therefore, the search for novel eco-friendly products is now crucial [6].

In addition, recent laws that progressively limit copper use for plant disease control in European countries have created an urgent need for alternative products [7]. These compounds include antibiotics [8], formulations containing acid and metal ions [9], natural products [10,11,12,13], as well as endophytic microorganisms that compete with *Xf* and downregulate its virulence. Additionally, specific phages capable of lysing *Xf* have also been explored [14]. The evaluation of the efficacy of natural products derived from plants is of particular interest for addressing negative effects on the environment and human health [15,16].

Antibacterial susceptibility testing (AST) is commonly employed to screen the *in vitro* antimicrobial activity of natural products. Various techniques, such as broth dilution (considered the gold standard for AST) [17], disk diffusion, and gradient-diffusion-based stripe tests for plate culture, are utilized for this purpose [18]. These methods rely on visual observations to determine the inhibition of strain growth, either through direct observation or with the aid of specialized optical instruments [19]. While these techniques are widely used, a significant drawback is either the significant time needed to prepare AST experiments or, in some cases, the requirement for large sample volumes. Alternative methods, such as mass spectrometry, real-time PCR, or microarray technology, have been reported [17]. However, these methods often face challenges such as insufficient sensitivity or the need for DNA extraction, as with real-time PCR. Additionally, some methods are still in the early stages of development and involve the use of expensive reagents and equipment, leading to high costs [17].

Recently, examples of AST using microfluidic channels have gained attention for their potential to improve and revolutionize AST through automation and miniaturization [17,20,21]. These systems offer key advantages, including the ability to culture bacteria in microscale environments that closely resemble *in vivo* biological conditions [22], enabling real-time monitoring of bacterial growth and morphology during the testing of potential antimicrobial compounds [23,24,25,26,27]. Furthermore, the many recently available techniques for channel fabrication [28,29,30] could facilitate the widespread adoption of microfluidics for AST. This result is a high-throughput assay with low sample consumption that integrates seamlessly with optical instruments for real-time observation of the culture [31].

In this study, a network of microfluidic channels for AST of natural products was investigated to address the growing global interest in natural bioactive compounds for managing bacterial diseases. Initially, the potential inhibition effects of natural compounds such as *Trametes versicolor* extract [32], clove oil extract, and Fossil^Ⓡ^ against *Xylella fastidiosa* subsp. *fastidiosa* (*Xff*) planktonic growth and biofilm formation were tested using the microdilution broth method, a widely employed and reliable AST technique regarded as the gold standard [20]. Subsequently, a microfluidic array, fine-tuned for cultivating *Xff*, was used to verify the inhibition effect of the aforementioned compounds. A quantitative and statistical analysis of the results on *Xff* planktonic inhibition and biofilm production was conducted, along with a comparison between the two AST techniques.

## 2. Results

### 2.1. Macrodilution Broth In Vitro Screening

In this study, macrodilution broth *in vitro* screening was the reference AST chosen to test the inhibition effect of the selected natural compounds against *Xff* and to evaluate their MIC (minimal inhibitory concentration). We tested both the ability of the substances to limit the planktonic growth of *Xff* after 3 and 6 days and the ability to inhibit biofilm formation after 6 days, expressed as a percentage of inhibition (Figure 1). The two substances, copper sulfate and *P. granatum* extract, were tested as chemical and natural controls, respectively, based on previously reported publications [13,33]. Regarding copper sulfate, nearly 100% inhibition of both *Xff* planktonic growth and biofilm formation was obtained at a concentration of 1.75 mg/mL (Figure 1, C2). Approximately 100% inhibition was also observed for copper sulfate concentrations of 1 and 2.5 mg/mL (C1 and C3). Since the results of the three copper sulfate concentrations—1 mg/mL, 1.75 mg/mL, and 2.5 mg/mL—are almost equivalent, the intermediate concentration (1.75 mg/mL) was used as the chemical control in all the experiments. The inhibitory effect of *P. granatum* extract was previously reported [13]; however, it was tested again using three different concentrations (C1 = 0.2 mg/mL, C2 = 2 mg/mL, and C3 = 20 mg/mL) under our experimental conditions. *P. granatum* extract exhibited an intense inhibition of planktonic growth at the C2 and C3 concentrations, with statistically significant results during all considered incubation times, as well as statistically significant inhibition of biofilm formation across all tested concentrations (Figure 1). This result, along with previous studies [13] that have demonstrated both the *in vivo* and *in vitro* efficacy of this extract against the pest, led us to select this compound for use as the natural control in our experiments. The strong efficacy of the tested extract appears to be due to its high content of total phenolic compounds, such as hydroxytyrosol glucoside, epicatechin, and gallic acid, which are often linked to potent antimicrobial activity [13].

Regarding the other natural compounds tested, the results from the macrodilution test indicate that after 3 and 6 days, both the planktonic growth of *Xff* (Figure 1a,b) and biofilm formation (Figure 1c) are inhibited by *T. versicolor* extract (C1 = 2.5 mg/mL, C2 = 5 mg/mL, and C3 = 10 mg/mL), clove oil (C1 = 0.01 mg/mL, C2 = 0.1 mg/mL, and C3 = 1 mg/mL), and Fossil^Ⓡ^ (C1 = 0.01 µg/mL, C2 = 0.1 µg/mL, and C3 = 1 µg/mL), as well as *P. granatum* extract and copper sulfate. However, the degree of inhibition and the statistically significant results vary based on the concentration tested, as shown in the graphs (Figure 1). The negative values observed at certain concentrations (Figure 1b,c) indicated a lower degree of inhibition, suggesting that the bacterium exhibits slight growth at these concentrations.

To determine the statistical significance of the results, an analysis of variance (ANOVA) was conducted to assess the differences between the means obtained for the inhibition of both planktonic growth and biofilm production. The ANOVA yielded extremely low *p*-values (2.04 × 10^−16^, 5.14 × 10^−13^, and 3.34 × 10^−11^ for planktonic growth after 3 and 6 days and biofilm, respectively), indicating a significant difference among the means derived from the macrodilution test for each substance and the respective concentrations.

A multiple-comparisons analysis was conducted between the *Xff* inhibition means obtained from the tested compounds compared to the inhibition value observed when copper sulfate was used. This analysis aimed to identify which substance and concentration significantly inhibit the planktonic growth of *Xff* and the formation of biofilm.

*T. versicolor* extract demonstrated antimicrobial activity at concentration C3 after 3 days, while C2 inhibition became significant only after 6 days of incubation. Clove oil exhibited significant antimicrobial activity after 3 days at concentrations C2 and C3; however, after 6 days, the inhibition by C2 was not statistically significant. In the case of Fossil^Ⓡ^, inhibition occurred only at concentration C3 after 3 and 6 days of incubation.

ANOVA and the multiple-comparisons analysis enabled the identification of concentrations that significantly inhibit *Xff* planktonic growth and biofilm formation and allowed for the determination of the MIC for each substance tested. The MIC values for each tested substance regarding the planktonic growth of the pathogen are as follows: 1 µg/mL for Fossil^Ⓡ^, 2 mg/mL for *P. granatum* extract, 1 mg/mL for clove oil, and 5 mg/mL for *T. versicolor* extract. Regarding biofilm inhibition, the MICs are as follows: 0.2 and 1 mg/mL for *P. granatum* extract and Fossil^Ⓡ^ and 1 and 10 mg/mL for clove oil and *T. versicolor* extract, respectively.

### 2.2. Microfluidic Channel In Vitro Screening

The device selected for this study is a microfluidic array with three straight channels made of glass and PDMS, with dimensions of 1 mm in width, 50 µm in depth, and 3 cm in length. The glass and PDMS materials provide biological compatibility with bacteria and allow for the rapid fabrication of multiple devices [34]. The conditions for growing *Xff* in the microfluidic channels were optimized. In particular, the effect of the initial concentration of the bacterial suspension, with an optical density measured at 600 nm (OD_600_), introduced into the channels and the flow rates of the medium required to support cell growth were examined.

#### 2.2.1. Cell Attachment on the Inner Channel Glass Surface

Bacterial suspensions in PD2 with OD_600_ values of 0.1, 0.5, and 0.8 were introduced into separate channels, each flowing at a rate of 0.25 µL/min. Additionally, one channel was dedicated solely to PD2 as a sterility test. This velocity ensured sufficient bacterial suspension entered the channels while preventing precipitation inside the syringe during injection, which was observed at lower velocities. Conversely, we noted that a higher flow rate would lead to excessive wastage of the bacterial suspension without enhancing bacterial adhesion to the channel interior. After 2.5 h, the flow was stopped, and PD2 was injected into all channels previously filled with bacterial suspension. This process was alternately performed at flow rates of 0.01, 0.05, and 0.1 µL/min overnight. This step was essential to rinse out cells that did not adhere to the channel surface. The variation of rinsing flow rates was tested to assess *Xff* adhesion to the microfluidic channel’s inner surface, with changes in the flow rate correlating to shear stress (S) [35]. In the channels used for these experiments, S was calculated to be 0.0035, 0.018, and 0.035 dyn/cm^2^ for flow rates of 0.01, 0.05, and 0.1 µL/min, respectively.

After the rinsing step, the channels were observed under the microscope, images were taken, and the size of the colonies was measured. The size of the colonies measured after the rinsing was set as t_0_ (t = time); subsequently, the bacteria were left to grow, and the channels were examined after 3 (t_3_) and 6 (t_6_) days. Only a few bacteria were visible on the channel surface when PD2 flowed through overnight at the rate of 0.05 and 0.1 µL/min, regardless of the concentrations of the bacterial suspensions used, while after 3 and 6 days, these channels were completely empty. However, when using a flow rate of 0.01 µL/min with overnight PD2 flow, some bacteria adhered to the channel’s inner surface; however, when bacterial suspensions with OD_600_ values of 0.1 and 0.5 were used, their distribution was non-uniform (Figure 2a,b). On the other hand, the channel filled with a bacterial suspension at an OD_600_ = 0.8 exhibited a uniform distribution of bacteria on the inner channel surface, with colony areas measuring approximately 15–25 µm^2^ (Figure 2c). Thus, a bacterial suspension with an OD_600_ = 0.8 and a PD2 flow velocity of 0.01 µL/min were applied in all the experiments.

#### 2.2.2. Cell Growing

The growth profile of the bacterial cells introduced into the channels using an OD_600_ = 0.8 was observed over 6 days. The growth profile of *Xff* in microfluidic channels is reported in Appendix A as the increment ratio of *Xff* colony areas as a function of time.

To assess the inhibitory effects of the substances previously examined using macrodilution broth *in vitro* screening, the evaluation was focused on determining the impact on both planktonic growth and biofilm formation of *Xff* by copper sulfate (C2 = 1.75 mg/mL), *P. granatum* extract (C2 = 2 mg/mL), *T. versicolor* extract (C3 = 10 mg/mL), clove oil (C3 = 1 mg/mL), and Fossil^Ⓡ^ (C3 = 1 µg/mL) within the microfluidic channel array, by using the MICs previously determined using the macrodilution broth method. For this purpose, a *Xff* suspension with an OD_600_ = 0.8 was introduced into the channels, followed by rinsing with PD2 for 24 h. Subsequently, PD2 solutions containing the products were introduced into the different channels. Time-lapse images of *Xff* cells were captured to monitor their growth over a 6-day experiment. After 3 and 6 days, the colony areas were measured, and the biofilm area was assessed following the introduction of a Crystal Violet solution (Figure 3 and Figure 4).

## 3. Discussion

The aim of this work was to evaluate the use of the microfluidic chip as an alternative valuable system for performing AST using select substances with a low environmental impact to inhibit *Xff* planktonic growth and biofilm formation. The substances tested in this study, being natural compounds, typically degrade more rapidly in the environment, reduce the risk of developing resistance, and have minimal toxic effects on non-target organisms, including beneficial insects, plants, and soil microbiota. Additionally, they often originate from renewable resources and are less likely to accumulate in the ecosystem compared to synthetic chemicals.

In this study, we investigated the antimicrobial potential of natural products, including *T. versicolor* extract [32] and clove oil [36,37], based on their documented efficacy against other pathogens. Additionally, we evaluated Fossil^Ⓡ^, a dual-action silicon–phosphite formulation that indirectly stimulates natural plant defenses, to test its potential direct effect on the pathogen. There are no reports about the inhibition effect of these compounds on Xff planktonic growth and biofilm formation.

The European Food Safety Authority stated the toxic effect of Cu^2+^ ions, which block enzyme reactions leading to the inactivation and denaturation of fungal spores and bacterial cells, when applied above the homeostatic range for microbes [33]. Furthermore, *P. granatum* extract, commonly known as pomegranate, was chosen as a reference of a natural compound that exhibits inhibitory effects against *Xff* planktonic growth and its biofilm formation [13]. Pomegranate extract was tested here under the same experimental conditions as *T. versicolor* extract, clove oil, and Fossil^Ⓡ^, and its antimicrobial efficacy was verified to be subsequently used in the microfluidic system.

It is interesting to note that the natural compounds tested in this study have antimicrobial efficacy similar to that of the well-known chemical agent of conventional control, i.e., copper sulfate. These achievements are promising for a future use of these compounds in the control of the diseases caused by the pathogens, and it would be interesting to test their *in planta* efficacy.

Building upon the MICs determined through macrodilution broth *in vitro* screening, we further investigated the inhibitory effects on *Xff* growth by the natural compounds within the microfluidic channel array.

Crucially, the use of microfluidic channels offers the advantage of minimizing reagent and cell culture volumes [17,38], a significant aspect when dealing with *Xff* given its propensity for slow *in vitro* growth (up to one week) conditions [39] compared to other bacterial plant pathogens. Microfluidic channels enable bacterial growth in confined compartments with minimal volumes (1–2 µL) that mimic the natural environment of plant xylem vessels and insect foreguts, providing a continuous flow condition for the growth of xylem-residing bacterial pathogens and ensuring the continuous support of nutrients [17]. The microfluidic platform can be easily integrated with a microscope, which allows for the measurement of colony growth areas and also facilitates the observation of bacterial morphology alterations under the influence of different substances, enabling continuous monitoring of colony growth under continuous flow conditions [38]. The chip array fabricated and applied for this work, consisting of three single channels, was selected for its easy design and fabrication, which does not require clean-room procedures, reducing the costs and time needed for fabrication and enabling its use in any laboratory. In this study, we established the optimized conditions for utilizing microfluidic channels as a platform to assess the inhibition effect of a substance against *Xff* planktonic growth and biofilm formation. Initial investigations focused on the adhesion of *Xff* to the inner channel wall, which would ensure the conditions for growing the bacteria under a constant flow of nutrients. Factors such as the shear stress (S) and the *Xff* concentration to be injected into the channels were considered. Given the small Reynolds number, microfluidic channels exhibit laminar flow that is characterized by a parabolic velocity profile, with the highest velocity at the center of the channel and the lowest at the channel wall. Consequently, the friction caused by the fluid’s tangential force is highest at the wall and lowest in the center of the channel. This tangential force, identified as shear stress, was observed to impact the adhesion of cells or tissues within the channel [40,41].

Additionally, we evaluated different OD_600_ values, providing good bacterial adhesion on the channel surface (Figure 2). Under our experimental conditions, considering both the uniformity of bacterial adhesion and bacterial growth within the channels, an OD_600_ of 0.8 was determined to be the necessary concentration for facilitating bacterial adhesion to the inner channel surface. However, to sustain *Xff* growth within the channel, the flow speed of PD2 was reduced to 0.01 µL/min. This flow rate corresponds to a lower shear stress compared to that utilized for anchoring the initial bacterial suspension onto the channel walls. This outcome implies that using a flow rate higher than 0.01 µL/min results in the rinsing out of bacteria from the inner surface [35], leaving only a few isolated cells that exhibit a slow growth rate (Figure 2a,b). As illustrated in Figure 3a, a flow rate of 0.01 µL/min permits both *Xff* planktonic growth and biofilm formation within 6 days.

The continuous flow of PD2 ensured a constant nutrient supply and facilitated the removal of metabolic waste. Under these conditions, bacterial cells were able to grow within the initial monolayer. This allowed for the direct monitoring of morphological variations in situ through simple bright-field microscopy, eliminating the need for a fluorescent-labeling protocol. Time-lapse images were employed to track their growth during the 6-day experiments. The growth curves of *Xff* cultured on the chip were compared to those in conventional test tubes, revealing a similar increment ratio of *Xff* colonies (see Appendix A).

Figure 3b–f illustrate that all the substances tested in the chip exhibit inhibition against *Xff* growth and its biofilm. Additionally, the increment ratio of *Xff* colonies was markedly lower in the channels where the substances were added to PD2 when compared to channels where only PD2 was used (refer to Appendix A).

An additional advantage of the channel array is that it allows for the observation of variations in *Xff* morphology under the constant flow of the products. As shown in Figure 3b,c, the inhibition effect due to the tested substance causes a decrease in the colony area when compared to that of the positive control (Figure 3a). Copper sulfate (Figure 3b) induces a noticeable change in the morphology of *Xff* within just 3 days, leading to the formation of small agglomerates that persist even after 6 days. These agglomerates may be the result of *Xff* lysis that left small residues on the channel surface. In contrast, for all other substances, after 3 days, *Xff* bacteria can still be identified individually, although with colonies significantly smaller compared to those formed under the constant flow of PD2. By the 6th day, a more substantial divergence in *Xff* morphology becomes apparent for each substance. Under the constant flow of *P. granatum* extract and Fossil^Ⓡ^, a few agglomerates like those observed for the copper sulfate emerge, while *T. versicolor* extract exhibits a further reduction in colony area and the formation of small filamentous shapes (Figure 3c,d). After 6 days of treatment with clove oil, the filamentation of *Xff* colonies becomes even more pronounced, highlighting the inhibition of bacterial agglomeration when compared to the positive control (Figure 3a,e). Filamentation of bacteria under antimicrobial treatment has been reported previously and is regarded as a survival mechanism [42].

A multiple-comparisons analysis between the means of *Xff* inhibition obtained by the natural compounds and that of copper sulfate (see Figure 4a–c) was performed. The asterisks in Figure 4 indicate that all the natural products tested in the microfluidic platform exhibit, at the MIC, an inhibition of *Xff* both on planktonic growth and biofilm formation, comparable to that of the copper sulfate and the natural compound *P. granatum* extract, thus confirming the results obtained using the macrodilution method. Both the macrodilution broth and microfluidic channel array techniques have unique strengths and limitations in AST. The macrodilution broth, as a standard technique, offers high accuracy in selecting compounds with antimicrobial activity. Nevertheless, microfluidic channel arrays have demonstrated comparable results, offering the advantage of reduced sample volumes and high throughput. In terms of speed, microfluidic channel arrays are less time consuming, enabling rapid experimental setup and result acquisition compared to the microdilution broth method [20,21]. This aspect is critical when assessing several substances. On the other hand, the macrodilution broth method still remains widely accessible and cost effective, and microfluidic technologies still require specialized equipment and expertise. In conclusion, although the traditional macrodilution broth method is still practiced, microfluidic technologies represent a valid alternative in terms of speed and precision. Despite this, further research and technological development are necessary to fully exploit the potential of microfluidic AST techniques to be adopted into routine laboratory activities.

## 4. Materials and Methods

### 4.1. Tested Compounds

*Trametes versicolor* extract, clove essential oil (clove oil), and Fossil^Ⓡ^ (Orion Future Technology, London, UK: https://www.orionft.com/products/fossil (accessed on 5 February 2025)) are the three natural substances that were tested in this study.

*T. versicolor* strain C used in this study was registered at CABI biosciences (UK) and deposited in the culture collection of the Department of Environmental Biology of Sapienza University of Rome as ITEM 11 [32]. *T. versicolor* extract was kindly provided by “Sapienza University of Rome”. The edible and non-toxic basidiomycete *T. versicolor* extract is able to produce bioactive substances [32], in particular exopolysaccharides and glycoprotein fractions. Clove oil was kindly provided by “Federico II” University of Naples (Valeria Giosafatto, unpublished). Clove oil is typically extracted using soxhlet extraction, a technique that involves immersing the clove material in a solvent and repeatedly cycling it through a heated chamber. The clove oil used in this study was obtained through soxhlet extraction with n-hexane for six hours, followed by rotary evaporation to remove the solvent. The final yield of clove oil was determined to be 18% ± 1. Clove oil boasts a rich blend of active constituents, including eugenol (4-allyl-2-methoxy phenol), a phenylpropanoid; eugenyl acetate, a monoterpene ester; and β-caryophyllene, a sesquiterpene. These compounds endow clove oil with potent antimicrobial and antioxidant properties, making it a valuable tool for extending food shelf life and inhibiting spoilage [36,43]. Commercial Fossil^Ⓡ^ is a resistance inductor and biostimulant product. Standard copper sulfate [44] and *Punica granatum* extract (kindly provided by the “CREA Research Centre for Engineering and Agro-Food Processing”, Italy [13], were used as chemical and natural controls, respectively.

### 4.2. Bacterium

*Xff* strain Temecula1 (NCPPB 4605) was grown in PD2 [8] agar medium for 7 days at 26 °C. The strain was maintained in the bacterial collection of the Research Centre for Plant Protection and Certification (CREA-DC, Rome, Italy), at −80 °C in phosphate-buffered saline (PBS) 1× containing 30% glycerol for long-term storage.

### 4.3. Minimum Inhibitory Concentration

AST based on macrodilution broth *in vitro* screening was used to determine the MIC. The MIC is the lowest concentration of an antibacterial agent that, under strictly controlled *in vitro* conditions, completely prevents visible growth of the test strain of an organism and defines *in vitro* levels of susceptibility or resistance of specific bacterial strains to an applied antibiotic [45].

For each substance, a stock solution was prepared as follows:For copper sulfate, 0.8 g was dissolved in 10 mL of sterile distilled water.For *P. granatum* extract, 2 g of extract was dissolved prior in 10 mL of sterile distilled water and stored overnight at −80 °C in glass vials; afterward, the solutions were freeze-dried at −40 °C for 2 days and stored at 4° C until use.*T. versicolor* cultural filtrate was obtained following a previously reported procedure [32] and split into aliquots of 10 mL, which were stored overnight at −80 °C in glass vials; afterward, the solutions were freeze-dried at −40°C for 2 days and stored at 4° C until use; the stock solutions were then prepared by dissolving 1 g of the lyophilized powder in sterile distilled water and sterilized using a 0.45 µm sterile filter.Clove oil was dissolved prior in dimethyl sulfoxide (DMSO): 20 mg of the oil was added to 250 μL of DMSO and subsequently sterilized using a 0.2 µm sterile filter.Fossil^Ⓡ^ stock solution was prepared by dissolving 1 g in 10 mL of sterile distilled water and subsequently sterilized by using a 0.2 µm sterile filter.

Prior to use, stock solutions were stored at 4 °C (*P. granatum* extract and *T. versicolor* extract) or alternatively at room temperature (copper sulfate, clove oil, and Fossil^Ⓡ^).

To perform the test, *Xff* pure culture was scraped off from the agar surface and resuspended in a sterile glass tube containing 2.5 mL of PD2 liquid broth and kept in a rotary shaker at 100 rpm and 26 °C, overnight. Subsequently, the absorbance at 600 nm of 500 μL of the bacterial suspension was spectrophotometrically measured by using the DeNovix Spectrophotometer DS-11 Fx+ (Denovix Inc., Wilmington, DE, USA) and adjusted to a concentration of approximately 10^7^ colony forming units (CFU)/mL (OD_600_ = 0.1) to be used as a starter inoculum.

Glass tubes containing 2.5 mL of PD2 broth or PD2 broth added with the substance were inoculated with 100 μL of the starter bacterial suspension to reach a final concentration of 10^6^ (CFU)/mL (OD_600_ = 0.01). For each substance used in this study, three concentrations were tested, as reported in Table 1. Three replications were performed for each dilution, and each experiment was carried out twice (n = 6).

The *in vitro* antibacterial efficacy of each compound on the planktonic state of *Xff* was evaluated by reading the optical density (OD_600_) of 500 μL of each concentration after 0, 3, and 6 days post-inoculation with the spectrophotometer. The biofilm production was also evaluated using the Crystal Violet assay after 6 days [46]. Each experiment was carried out twice. The percentage of planktonic and biofilm inhibition by the substances was calculated by using the following equation [12]: ([1 − (ODt_600_ − ODc_600_)]/ODc_600_) × 100; here, ODc_600_ and ODt_600_ are the optical densities of the suspended cells measured initially (t = 0) at time t, respectively.

### 4.4. Microfluidic Channel In Vitro Screening

The microfluidic network applied for this study is illustrated in Figure 5 and Appendix A and consists of a microscope glass slide (Prestige^®^; 2.4 × 6 cm) having a thickness of 0.13–0.17 mm bonded to a polydimethylsiloxane (PDMS) slab having three channels with separated inlets and outlets of the following dimensions: 3 cm length, 1 mm width, and 0.05 mm depth (for the fabrication procedure, see Appendix A).

*Xff* was grown on PD2 agar for 4/5 days at 26 °C; afterward, the bacterial cells were scraped from the agar surface, suspended in PD2 broth (OD_600_ = 0.8), and dispersed via pipetting. In each experiment, two substances were tested by using two devices. Each substance was tested twice. In order to perform the experiment, three of the channels were filled with bacterial suspension in PD2 and three with PD2 only (sterility controls) at a flow rate of 0.25 µL/min for 2.5 h, and then the flow was stopped for 30 min. Subsequently, the syringes and inlets used for bacterial injections were changed, and all the channels were rinsed with PD2 broth at 0.01 µL/min overnight to remove floating bacteria not adhered to the glass–PDMS channel inner surface. After this rinsing step, the channels were observed using the optical microscope. Of the three channels previously treated with the bacterial suspension, two were filled with the PD2 solution with the added substance and one with PD2 broth only using a flow rate of 0.01 µL/min, respectively. The second device, initially filled only with PD2, was used as the sterility control: two channels were filled with PD2 with the added substances and one with PD2 broth only, maintaining a flow rate of 0.01 µL/min, respectively. The growth of *Xff* in the microfluidic array was evaluated after 3 and 6 days using the optical microscope.

The formation of the biofilm was evaluated after 6 days by flowing a solution of 0.1% Crystal Violet at a flow rate of 0.1 µL/min for 30 min. Subsequently, the biofilm area was evaluated with the optical microscope.

### 4.5. Optical Image Acquisition and Analysis

Time-lapse images were recorded via observation through the cover glass slide using an optical microscope equipped with a 40× objective lens (Zeiss Axioskop 2 Phase Contrast Brightfield Microscope, Carl Zeiss S.p.A., Milan, Italy; Nikon digital sight DS-F1, Nikon, Tokyo, Japan). All the images were processed using ImageJ 1.52a free software. The images were first converted to an 8-bit type, the background was subtracted, and auto-contrast was applied to bright-field images to give clearly defined cell edges. The areas of each colony were calculated using the analyzing particle function in ImageJ (the circularity parameter of 0–1 and particle size from 0 to infinity), with the results being given in micrometers. The micrometer scale was settled by measuring the number of pixels corresponding to the scale bar of the microscope image. In particular, we overlapped the “straight” bar option of ImageJ with the scale bar on the image and subsequently used the option “set-scale” to calculate the micrometer corresponding to the image pixels. For each experiment, the channel was divided into three areas of 1 cm, starting from the inlet hole, and colonies of nine images for the areas were analyzed considering the colonies adhered to the glass inner side of the channel.

The percentage of inhibition (I) was calculated by using the following equation [12]: I = ([1 − (S_t_ − S_0_)]/S_0_) × 100; here, S_0_ and S_t_ are the bacterial colony areas measured in the channels initially (t = 0) and at time t, respectively.

The shear stress (dyn/cm^2^) was calculated using the following formula [47]: S = 6 µQ/wh^2^; here, µ is the fluid viscosity of the solution, Q is the volumetric flow rate (cm^3^/s), w is the channel width, and h is the channel depth. In this experiment, the viscosity of the liquid medium was considered as that of water, which is 0.0089 dyn s/cm^2^.

One-way analysis of variance (one-way ANOVA) was performed using MATLAB 2018 software and was used to compare significantly different means of the *Xff* planktonic and biofilm inhibition obtained after treatment with the products.

## 5. Conclusions

Based on the results achieved in this study, through both the AST methods, we showed that *T. versicolor* extract, clove oil, and Fossil^Ⓡ^ exhibit inhibition against *Xff* planktonic growth and biofilm formation, as well as inhibition by *P. granatum* extract was confirmed under our experimental conditions. Also, we demonstrated that microfluidic technology is a valuable means to cultivate *Xff* under continuous flow. In these devices, AST of multiple substances can be performed simultaneously using a small volume of bacterial cells suspension. Microfluidic-based AST offers several advantages over traditional methods. It improves efficiency in terms of both time and cost. The experiment setup is faster, since it only requires flowing *Xff* into the microfluidic channels, eliminating the need for multiple pipetting steps. Additionally, real-time monitoring of bacterial inhibition through microscope imaging automates the analysis, including colony area calculation, and removes the need for optical density measurements, as is required in test tube-based methods.

Another advantage is the significant reduction in reagent consumption and waste. The small dimensions of the microfluidic channels minimize the volume of reagents required, reducing costs and enabling high-throughput testing with multiple substance concentrations within a compact space.

Real-time microscopy further enhances the process by providing detailed bacterial growth patterns and morphology changes. In the case of *Xff*, this technique can also contribute to understanding the mechanisms behind bacterial inhibition.

In conclusion, microfluidic AST has the potential to become a universal method, overcoming certain limitations related to microfluidic fabrication, which is not always straightforward. The rapid diagnostic and automation potential of microfluidic AST are paving the way for wider adoption in laboratories.

## Figures and Tables

**Figure 1 plants-14-00872-f001:**
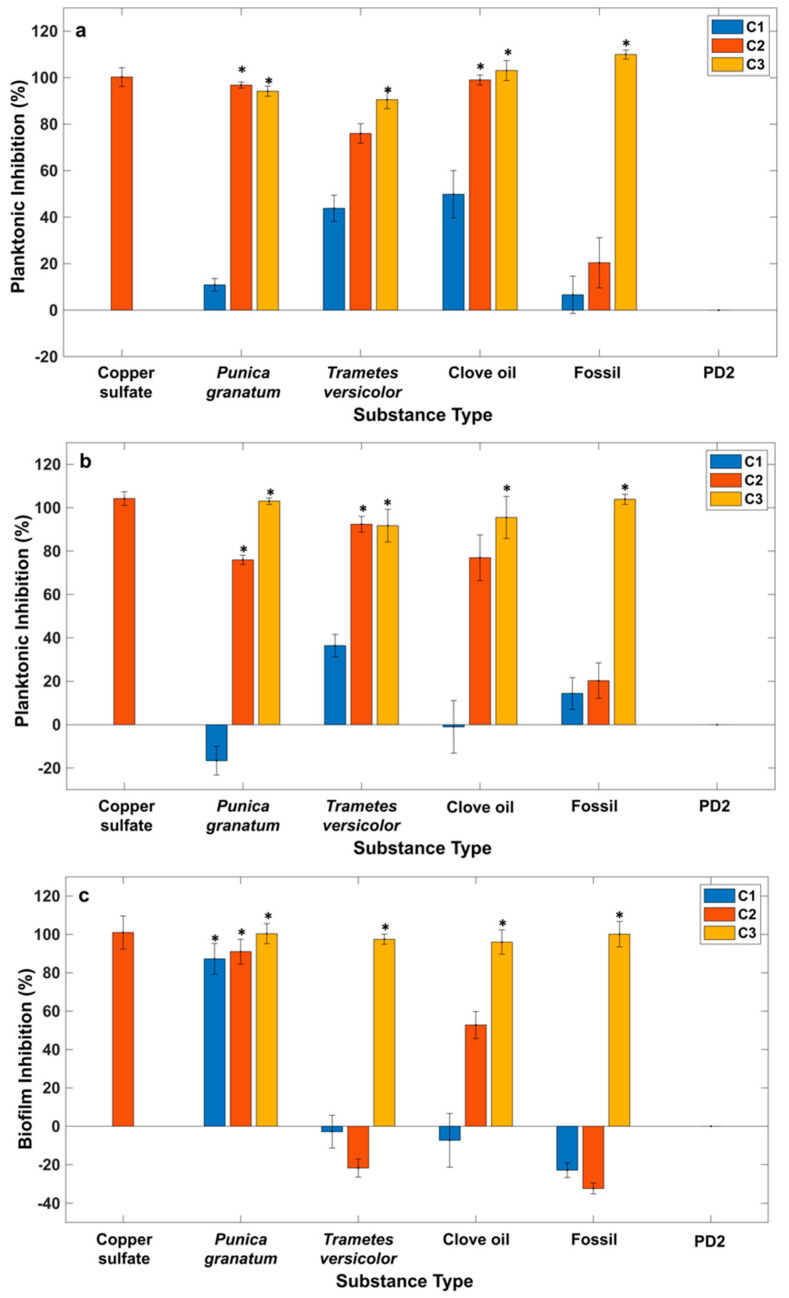
Bar chart representation of antimicrobial activities of the analyzed substances against *Xylella fastidiosa* subsp. *fastidiosa* after (**a**) 3 and (**b**) 6 days and (**c**) biofilm inhibition after 6 days using macrodilution broth *in vitro* screening: copper sulfate (C2 = 1.75 mg/mL), *Punica granatum* extract (C1 = 0.2 mg/mL, C2 = 2 mg/mL, and C3 = 20 mg/mL), *Trametes versicolor* extract (C1 = 2.5 mg/mL, C2 = 5 mg/mL, and C3 = 10 mg/mL), clove oil (C1 = 0.01 mg/mL, C2 = 0.1 mg/mL, and C3 = 1 mg/mL), Fossil^Ⓡ^ (C1 = 0.01 µg/mL, C2 = 0.1 µg/mL, and C3 = 1 µg/mL), and PD2 = sterility control. The asterisks indicate statistically significant antimicrobial activity compared to that of copper sulfate; error bars correspond to the standard deviation of n = 6.

**Figure 2 plants-14-00872-f002:**
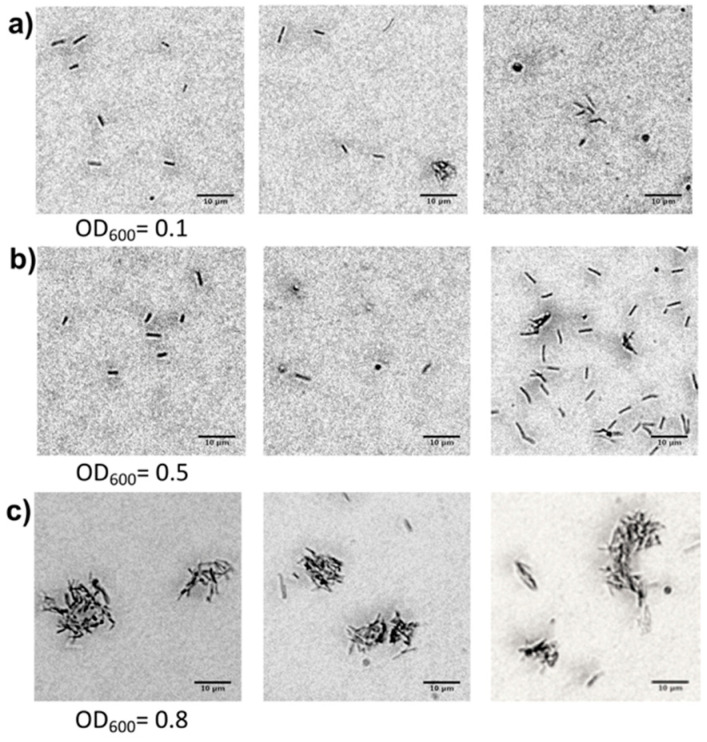
Microscope images of *Xylella fastidiosa* subsp. *fastidiosa* bacterial cells adhered to the inner surface of a microfluidic channel after 24 h of rinsing with PD2 at the flow rate of 0.01 µL/min (S = 0.0036 dyn/cm^2^). The bacterial suspension used had OD_600_ values of (**a**) 0.1, (**b**) 0.5, and (**c**) 0.8.

**Figure 3 plants-14-00872-f003:**
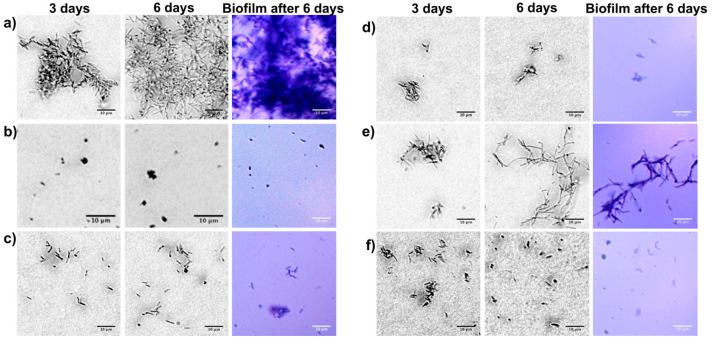
Microscope images of *Xylella fastidiosa* subsp. *fastidiosa* bacterial cells and biofilm under a constant flow of (**a**) PD2, (**b**) copper sulfate (1.75 mg/mL), (**c**) *Punica granatum* extract (2 mg/mL), (**d**) *Trametes versicolor* extract (10 mg/mL), (**e**) clove oil (1 mg/mL), and (**f**) Fossil^Ⓡ^ (1 µg/mL).

**Figure 4 plants-14-00872-f004:**
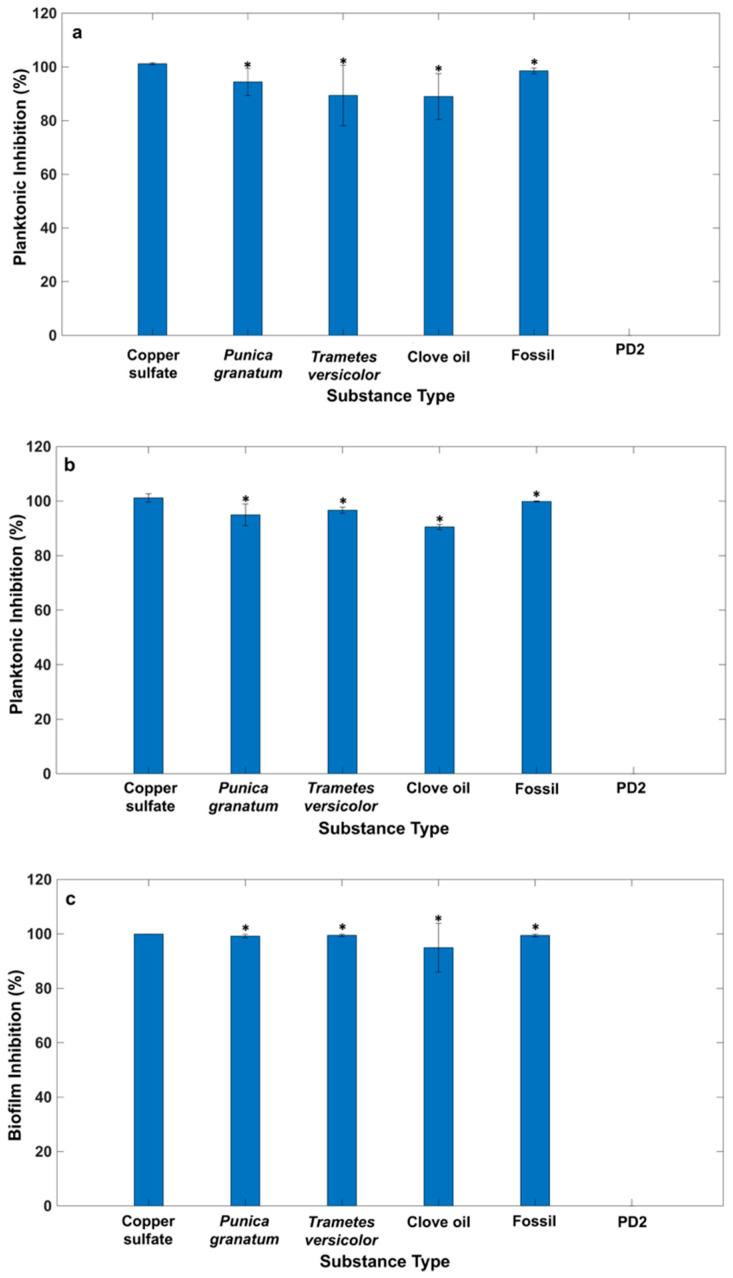
Bar chart representation of antimicrobial activities of the analyzed substances in the microfluidic array against *Xylella fastidiosa* subsp. *fastidiosa* after (**a**) 3 days and (**b**) 6 days and (**c**) after 6 days for the biofilm inhibition assay. Each substance was tested using the following concentrations: copper sulfate 1.75 mg/mL (C2), *Punica granatum* extract 2 mg/mL (C2), *Trametes versicolor* extract 10 mg/mL (C3), clove oil 1 mg/mL (C3), Fossil^Ⓡ^ 1 µg/mL (C3), and PD2 = sterility control. The asterisks indicate statistically significant antimicrobial activity compared to that of copper sulfate.

**Figure 5 plants-14-00872-f005:**
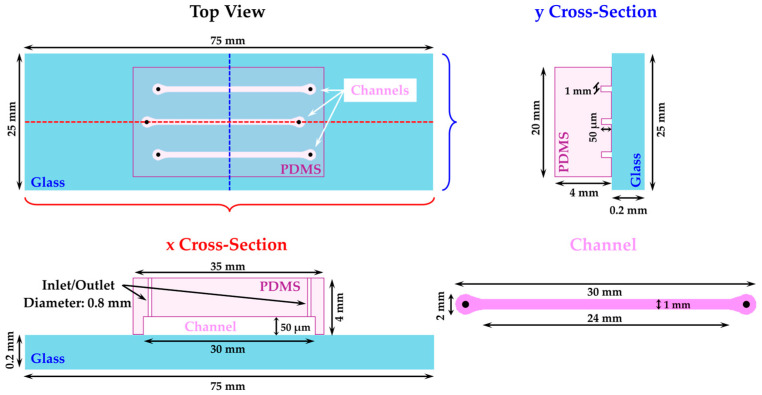
Schematic representation of the microfluidic array: top view, x cross-section, y cross-section, and channel.

**Table 1 plants-14-00872-t001:** List of the different compounds tested and the relative concentrations.

Substance	Concentration 1 (C1)	Concentration 2 (C2)	Concentration 3 (C3)
Copper sulfate	1 mg/mL	1.75 mg/mL	2.5 mg/mL
*Punica granatum* extract *	0.2 mg/mL	2 mg/mL	20 mg/mL
*Trametes versicolor* extract **	2.5 mg/mL	5 mg/mL	10 mg/mL
Clove oil	0.01 mg/mL	0.1 mg/m	1 mg/mL
Fossil^Ⓡ^	0.01 μg/mL	0.1 μg/mL	1 μg/mL

* *Punica granatum* extract was obtained from pomegranate fruit; ** *Trametes versicolor* extract was obtained from edible and non-toxic basidiomycete *T. versicolor.*

## Data Availability

The authors declare that the data supporting the findings of this study are available within the paper and its Supplementary Information files. Should any raw data files be needed in another format, they are available upon reasonable request. Source data are provided for this paper at the CREA-DC database.

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
