# Peer review of "Microfluidic Array Enables Rapid Testing of Natural Compounds Against Xylella fastidiosa"

_plants, 2025, doi:10.3390/plants14060872_

Round 1

Reviewer 1 Report

Comments and Suggestions for Authors

This manuscript is focalized on the creation of the novel method of an antibacterial susceptibility testing (AST) based on microfluidic channels.
The study investigates the in vitro antibacterial effects of natural compounds, including Trametes versicolor extract, clove essential oil and the resistance inducer Fossil®, against X. fastidiosa subsp. fastidiosa
Here are some comments on each chapter of the paper:
Introduction.
In the introduction section please provide detailed explanations of why Trametes versicolor and P. granatum extracts were used to develop the methodology?

Methods:
What active compounds with antimicrobial activity are contained in your extracts? Please, include short description in methods

What strain was used? Provide a comparison with other commercial preparations based on this species.

Why was Punica granatum extract used as a natural control? What components does this extract contain?

In conclusion, it is worth to describe in detail the advantage of this test in comparison with the other available? Will this method be a universal?
How can different biopesticides containing different active ingredients and their amounts be compared with each other?

Comments on the Quality of English Language

The English could be improved to more clearly express the research.

Author Response

Reviewer 1

This manuscript is focalized on the creation of the novel method of an antibacterial susceptibility testing (AST) based on microfluidic channels.
The study investigates the in vitro antibacterial effects of natural compounds, including Trametes versicolor extract, clove essential oil and the resistance inducer Fossil®, against X. fastidiosa subsp. fastidiosa Here are some comments on each chapter of the paper:

Comment n. 1 Introduction.
In the introduction section please provide detailed explanations of why Trametes versicolor and P. granatum extracts were used to develop the methodology?

Answer

We thank the reviewer for the comment.

  1. granatum extract was not used for the development of the method; P. granatum was used as positive control to test other substances with unknown efficacy due to its proven efficacy against the bacterium (see reference n.13). Therefore, about Trametes versicolor, clove oil and Fossil we added the following sentence:

Line 76: In this study, a network of microfluidic channels for antimicrobial sensitivity testing (AST) of natural products was investigated to address the growing global interest in natural bioactive compounds for managing bacterial diseases.

Comment n. 2 Methods: What active compounds with antimicrobial activity are contained in your extracts? Please, include short description in methods

Answer

We thank the reviewer for the comment.
In line 369 we added: “The edible and non-toxic basidiomycete Trametes versicolor extract is able to produce bioactive substances [32], in particular exopolysaccharides and glycoprotein fractions.”

From line 372 we described the extraction methods of clove oil and its main components.

Comment n. 3 What strain was used? Provide a comparison with other commercial preparations based on this species.

Answer

We thank the reviewer for the comment.

Regarding the strains: We added this sentence in line 366: “T. versicolor strain C used in this study was registered at CABI biosciences (UK) and deposited in the culture collection of Department of Environmental Biology of Sapienza University of Rome as ITEM 117.”

Regarding the commercial preparation:

The efficacy in vitro of P. Granatum was already reported, see reference n. 13. Pomegranate extract has been researched as a natural alternative to synthetic pesticides (Plants 2021, 10, 453) and its use is still under investigation. In this work, P. granatum is exclusively used as control, as reported in line 101 (see also Reviewer Comments 1 and 4). The P granatum extract provided by Dr. Rongai was already tested with Xf see reference n13. Moreover, there are several commercial preparations based on Trametes versicolor (Turkey Tail), mainly used in medicine, agriculture, and biotechnology. These include extracts, powders, and supplements with applications in human health, plant protection, and bioremediation. In this work, we used the extract prepared by the lab of Prof. Reverberi at Sapienza university of Rome, the details about the active compounds are included in as previously reported publication see reference n. 32. At the best of our knowledge, there are not previously reports about inhibition of T. versicolor versus Xf.

Clove oil: the active compounds present in the extract are reported in materials and methods in line 372. Clove oil is an essential oil extracted from the dried flower buds of the clove tree (Syzygium aromaticum). It is rich in eugenol, a compound with powerful antimicrobial, analgesic, and antioxidant properties. In agriculture is used as Biopesticide – Natural insect repellent against mosquitoes, termites, and other pests and Antifungal treatment for crops – Used in organic farming to combat fungal diseases see references n. 36 and 37. Some companies commercialize clove oil-based products: EcoSmart® Insect Killer Spray – For natural pest control in organic farming. At the best of our knowledge, clove oil was never tested with Xf.

Fossil is a commercial available products as described in line 256.

To clarify all these points, we inserted the references about clove oil in the discussion and modified the sentence in line 254: “In this study, we investigated the antimicrobial potential of natural products including T. versicolor [32] and clove oil extracts [36,37], based on their documented efficacy against other pathogens. Additionally, we evaluated FossilⓇ, a dual-action silicon-phosphite formulation that indirectly stimulates natural plant defenses, to test its potential direct effect on the pathogen. There are no reports about the inhibition effect of these compounds on Xff planktonic growth and biofilm formation.”

Comment n. 4 Why was Punica granatum extract used as a natural control? What components does this extract contain?

Answer

We thank the reviewer for the comment.

We added this sentence in line 107: This result, along with previous studies [13] that have demonstrated both the in vivo and in vitro efficacy of this extract against the pest, led us to select this compound for natural control in our experiments. The strong efficacy of the tested extract appears to be due to its high content of total phenolic compounds, such as hydroxytyrosol glucoside, epicatechin, and gallic acid, which are often linked to potent antimicrobial activity [13].

Comment n. 5 In conclusion, is it worth to describe in detail the advantage of this test in comparison with the other available? Will this method be a universal?

Answer

Thank you for your comment. In order to clarify this point, we added the “Conclusions” in the text, as follows:

Microfluidic-based antimicrobial susceptibility testing (AST) offers several advantages over traditional methods. It improves efficiency in terms of both time and cost. The experiment setup is faster since it only requires flowing Xff into the microfluidic channels, eliminating the need for multiple pipetting steps. Additionally, real-time monitoring of bacterial inhibition through microscope imaging automates the analysis, including colony area calculation, and removes the need for optical density (OD) measurements as required in test tube-based methods.

Another advantage is the significant reduction in reagent consumption and waste. The small dimensions of the microfluidic channels minimize the volume of reagents required, reducing costs and enabling high-throughput testing with multiple substance concentrations within a compact space.

Real-time microscopy further enhances the process by providing detailed bacterial growth patterns and morphology changes. In the case of Xff, this technique can also contribute to understanding the mechanisms behind bacterial inhibition.

In conclusion, microfluidic AST has the potential to become a universal method, overcoming certain limitations related to microfluidic fabrication, which is not always straightforward. Its rapid diagnostic capabilities and automation potential are paving the way for wider adoption in laboratories.

Comment n. 6

How can different biopesticides containing different active ingredients and their amounts be compared with each other?

Answer

Thank you for your comment.

In this work we compared biopesticides with different active ingredients with each other by evaluating their antibacterial efficacy under standardized in vitro conditions, such as culture conditions, positive and negative controls, replications and statistical analysis of the tests to assess differences in MIC values.

English was improved by a native speaker.

Reviewer 2 Report

Comments and Suggestions for Authors

The manuscript is well-written and clear (mostly), but I have a comment regarding the MIC assay. It is not clear to me haw the dilutions were preparedand how the concentratrations for further testing were selected. I have included my comments in the text.

Author Response

Reviewer 2

The manuscript is well-written and clear (mostly), but I have a comment regarding the MIC assay. It is not clear to me how the dilutions were prepared and how the concentrations for further testing were selected. I have included my comments in the text.

Comment n. 1

Why are the results for MIC values not shown? In what concentration range did you perform the MIC assay?(pg3) comparing to?

Answer.

Thank you for the question:

In line 157-161, the MIC values are reported: “The MIC values for each tested substance regarding the planktonic growth of the pathogen are as follows: 1 µg/mL for Fossil®, 2 mg/mL for and P. granatum, and 1mg/ mL for clove oil and 5 mg/mL for T. versicolor. Regarding biofilm inhibition, the MICs are as follows: 0.2 and 1 mg/mL for P. granatum and Fossil®, and 1 and 10 mg/mL for clove oil and T. versicolor, respectively.”

In Table 1 of “materials and methods”: we reported the concentrations used, for each substance, to identify the MIC.

In line 141 we modified the sentence: “A multiple comparison analysis was conducted between the Xff inhibition means obtained from the tested compounds compared to the inhibition value observed when copper sulfate was used”

Comment n.2

Please explain two MIC concentrations. Minimal inhibitory concentration (MIC) defines in vitro levels of susceptibility or resistance of specific bacterial strains to applied antibiotic or test substances.

Answer

Thank you for the suggestion:

We introduced the following sentence with relative reference in line 392: MIC is the lowest concentration of an antibacterial agent which, under strictly controlled in vitro conditions, completely prevents visible growth of the test strain of an organism and defines in vitro levels of susceptibility or resistance of specific bacterial strains to applied antibiotic [45].

Comment n.3

The results of bacterial growth for untreated bacteria, is it Tube? Please marked in the Figure.

Answer.

Thank you for the question: The experiment with untreated bacteria was performed both in tubes and in microfluidics, as show in Figure S2 of supplementary material and described in the “Materials and Methods” section. See also Figure 3a.

Figure S2 was marked in the caption.

Comment n.4

What was the concentration range. Usually MIC assay is performed with two-fold dilution of test substances.

Answer

Thank you for the comment. We used 10-fold dilution as reported previously, see reference n. 46.

Reviewer 3 Report

Comments and Suggestions for Authors

Title:      

Microfluidic array enables rapid testing of natural compounds against Xylella fastidiosa

Authors

Francesca Costantini1, Erica Cesari, Nicola Lovecchio, Marco Scortichini, Valeria Scala, Stefania Loreti1 and Nicoletta Pucci

The manuscript focus on investigating the antimicrobial potential of natural products, including  T. versicolor and clove oil extracts, based on their documented efficacy against other pathogens. Additionally, it was evaluated FossilⓇ, a dual-action silicon-phosphite formulation that indirectly stimulates natural plant defenses, to test its potential direct effect on the pathogen.

  1. “The device selected for this study is a microfluidic array with three straight channels made by glass and PDMS, with dimensions of 1 mm in width, 50 μm in depth, and 3 cm  in length.”

Please draw the sketch map of the device.

  1. “Building upon the MICs determined through the macrodilution broth in vitro screening, we further investigated the inhibitory effects on Xff growth by the natural compounds within the microfluidic channel array.”
  • Please provide the minimum inhibitory concentration (MIC) of natural compounds against Xylella fastidiosa through the macrodilution broth in vitro screening and the microfluidic channel array.
  • Compare the different above.

Comments on the Quality of English Language

The English could be improved to more clearly express the research.

Author Response

Reviewer 3

The manuscript focus on investigating the antimicrobial potential of natural products, including T. versicolor and clove oil extracts, based on their documented efficacy against other pathogens. Additionally, it was evaluated FossilⓇ, a dual-action silicon-phosphite formulation that indirectly stimulates natural plant defenses, to test its potential direct effect on the pathogen.

Comment n. 1

  1. “The device selected for this study is a microfluidic array with three straight channels made by glass and PDMS, with dimensions of 1 mm in width, 50 μm in depth, and 3 cm  in length.”

Please draw the sketch map of the device.

Answer.

Thank you for the comment: we added the scheme of the microfluidic array, see Figure 5

Comment n. 2

“Building upon the MICs determined through the macrodilution broth in vitro screening, we further investigated the inhibitory effects on Xff growth by the natural compounds within the microfluidic channel array.”

  • Please provide the minimum inhibitory concentration (MIC) of natural compounds against Xylella fastidiosa through the macrodilution broth in vitro screening and the microfluidic channel array.
  • Compare the different above.

Answer

Thank you for the comment. The MIC from the standard macrodilution in vitro test is reported in lines 157-161. The same concentrations were applied in the microfluidic chip experiments to demonstrate that the on-chip procedure could be effectively used  for antimicrobial susceptibility testing (AST). The obtained results confirm the possible future application of the microfluidic array for AST as standard method for AST. As a matter of fact, the statistical analysis (ANOVA) of the inhibition results obtained with the microfluidic array confirmed the significance of the microfluidic array data (see Figures 1 and 4). Also, we have updated the Conclusions section to summarize the advantages of the microfluidic approach.

English was improved by a native speaker.

Round 2

Reviewer 1 Report

Comments and Suggestions for Authors

Accept in present form

Author Response

Thank you for reading and comment on this work.

Reviewer 3 Report

Comments and Suggestions for Authors

The manuscript can be accepted now.

Comments on the Quality of English Language

The English could be improved to more clearly express the research.

Author Response

Thank you for reading and comment on this work.

The English was proofread by a native speaker.